# Accelerated partner therapy (APT) partner notification for people with *Chlamydia trachomatis*: protocol for the Limiting Undetected Sexually Transmitted infections to RedUce Morbidity (LUSTRUM) APT cross-over cluster randomised controlled trial

Claudia S Estcourt [ID],[1] Alison R Howarth,[2] Andrew Copas,[2] Nicola Low [ID],[3] Fiona Mapp [ID],[2] Melvina Woode Owusu,[2] Paul Flowers,[4] Tracy Roberts,[5] Catherine H Mercer,[2] Sonali Wayal,[2,6] Merle Symonds,[7] Rak Nandwani,[8] John Saunders,[2,9] Anne M Johnson,[2] Maria Pothoulaki,[1] Christian Althaus,[3] Karen Pickering,[5] Tamsin McKinnon,[2] Susannah Brice,[10] Alex Comer,[10] Anna Tostevin,[2] Chidubem (Duby) Ogwulu,[5] Gabriele Vojt,[1] Jackie A Cassell [ID] [11]

**Correspondence to**
Dr Claudia S Estcourt;
claudia.estcourt@gcu.ac.uk

## ABSTRACT

**Introduction** Partner notification (PN) is a process aiming to identify, test and treat the sex partners of people (index patients) with sexually transmitted infections (STIs). Accelerated partner therapy (APT) is a PN method whereby healthcare professionals assess sex partners, by telephone consultation, before giving the index patient antibiotics and STI self-sampling kits to deliver to their sex partner(s). The Limiting Undetected Sexually Transmitted infections to RedUce Morbidity programme aims to determine the effectiveness of APT in heterosexual women and men with chlamydia and determine whether APT could affect *Chlamydia trachomatis* transmission at population level.

**Methods and analysis** This protocol describes a cross-over cluster randomised controlled trial of APT, offered as an additional PN method, compared with standard PN. The trial is accompanied by an economic evaluation, transmission dynamic modelling and a qualitative process evaluation involving patients, partners and healthcare professionals. Clusters are 17 sexual health clinics in areas of England and Scotland with contrasting patient demographics. We will recruit 5440 heterosexual women and men with chlamydia, aged ≥16 years.
The primary outcome is the proportion of index patients testing positive for *C. trachomatis* 12-16 weeks after the PN consultation. Secondary outcomes include: proportion of sex partners treated; cost effectiveness; model-predicted chlamydia prevalence; experiences of APT. The primary outcome analysis will be by intention-to-treat, fitting random effects logistic regression models that account for clustering of index patients within clinics and trial periods. The transmission dynamic model will be used to predict change in chlamydia prevalence following APT. The economic evaluation will

use mathematical modelling outputs, taking a health service perspective. Qualitative data will be analysed using interpretative phenomenological analysis and framework analysis.

**Ethics and dissemination** This protocol received ethical approval from London—Chelsea Research Ethics Committee (18/LO/0773). Findings will be published with open access licences.

**Trial registration number** ISRCTN15996256.

## Strengths and limitations of this study

► The trial will evaluate accelerated partner therapy (APT), a novel approach to partner notification (PN), designed to facilitate and accelerate testing and treatment of sex partners of people with chlamydia.
► The primary outcome, chlamydia positivity, is objectively measured and clinically relevant, given its negative impact on both individual health outcomes and NHS (National Health Service) costs.
► This clinic-wide low-risk intervention has been granted ethical approval at the service-level without need for individual consent and complies with GDPR (General Data Protection Regulation).
► The pragmatic trial design will ensure that the effectiveness of APT is evaluated under real-life clinical conditions.
► An integral process evaluation will capture any variations in the operationalisation of PN and APT at the local level and enable optimisation of APT procedures in any future roll-out.

## INTRODUCTION

Partner notification (PN) is the process of identifying, testing and treating sex partners of a person with a sexually transmitted infection (STI).[1] PN is a key element of STI control on several levels.[2] It should benefit the individual diagnosed with the STI (the index patient) by preventing re-infection, the sex partner who might be the source of infection or could transmit undiagnosed infections to new sexual partners, and it should help to reduce spread of STIs in sexual networks and populations.

STIs are a major public health concern. The sexually transmitted pathogen *Chlamydia trachomatis* causes chlamydia infection, the most commonly reported bacterial STI in Britain,[3–5] with 218 095 diagnosed cases in England in 2018.[3] Untreated chlamydial infection can lead to pelvic inflammatory disease (PID), infertility, ectopic pregnancy and chronic pelvic pain in women.[6] In prospective studies with active follow-up, about 20% of women have a repeat diagnosis of chlamydia infection in the year after treatment.[7–9]

Chlamydia screening programmes aim to reduce the occurrence of complications by detecting and treating infections. For example, the incidence of diagnosed chlamydia in England has increased because of increases in both testing and transmission.[3 10–12] Chlamydia control activities in England include a National Chlamydia Screening Programme, which offers chlamydia testing to sexually active adults aged under 25 years, who accounted for 61% of diagnosed chlamydia infections in 2018.[13] The Public Health Outcomes Framework in England sets targets to increase chlamydia diagnosis in young men and women.[14] However a mathematical modelling study suggested that improving PN outcomes for chlamydia would be more cost effective than increasing the coverage of chlamydia testing,[2] emphasising the importance of optimising the effectiveness of PN whatever the coverage of testing.

Like many other bacterial STIs, chlamydia does not induce lasting immunity after antibiotic treatment, and this represents a particular challenge for STI control efforts. Repeated infection after treatment can result from re-infection from an untreated partner, indicating failed PN, or from a new infection from a new partner. Alternatively, infection might persist if antibiotic treatment is not effective for any reason. It is usually not possible to determine the reason for the repeat positive test (re-infection, new infection or treatment failure). In a mathematical modelling study, the peak incidence of repeated infections was estimated at 2–5 months after treatment.[15]

Given the importance of PN in the control of STIs such as chlamydia, there is a need to optimise sexual health services' support for effective PN. Currently, healthcare professionals in British sexual health services often struggle to meet modest targets.[16–18] The PN process is challenging both for patients who may face barriers to informing sex partner(s) about the STI, and to services where trained staff need time to elicit sensitive information from patients in order to support PN.

Offering patients alternatives, so that they can choose the most acceptable PN method—which might differ between sex partners—is considered optimal practice.[19] For people with chlamydia, PN is most often performed through patient referral,[20 21] in which a healthcare professional advises the person with a diagnosed infection (the index patient) to inform their sex partner(s) of the need for testing and treatment and to refer them to a sexual health service (simple patient referral). This basic advice can be supplemented by information, in the form of written leaflets or website addresses, for the index patient to give to their partner(s) (enhanced patient referral). Expedited partner therapy (EPT) is a strategic approach to enhanced patient referral, widely adopted in USA, in which a healthcare professional gives the index patient antibiotics or a prescription for their partner(s).[22] A systematic review found that EPT results in lower proportions of index cases with repeated curable STIs than simple patient referral,[16] and a US study demonstrated decreases in chlamydia positivity and gonorrhoea incidence at the population level.[23] However, EPT, as now widely practised in USA, does not require a consultation with the sex partner and does not comply with UK prescribing guidance,[24] so cannot be implemented in UK.

We therefore developed accelerated partner therapy (APT) as an adaptation of EPT to speed up the enhanced patient referral process.[25–27] APT complies with UK prescribing guidance since a healthcare professional assesses the appropriateness of the prescribed antibiotics for the partner.[28] In brief, the healthcare professional performs a telephone consultation with the sex partner in private during the index patient's clinic attendance. If medically safe, the index patient receives an APT pack, containing antibiotics and self-sampling kits for STI and HIV to deliver to their sex partner(s), or the clinic may post the APT pack to the sex partner(s). In pilot studies, APT resulted in faster sex partner treatment and greater overall numbers of sex partners treated, when compared with standard PN, but lower levels of testing for HIV and other STIs, when offered without HIV testing as part of the pack.[25 26]

In the trial described here, we aimed to compare strategies for PN, specifically APT with routine PN approaches as currently practised in UK. As part of the Limiting Undetected Sexually Transmitted infections to RedUce Morbidity (LUSTRUM) programme (lustrum.org.uk), we designed a randomised controlled trial (RCT) to determine the effects of APT for people with *C. trachomatis* on biological, as well as patient-reported clinical outcomes. Specific objectives are to determine: (1) the effect of APT on the proportion of index patients who test positive for chlamydia 12–16 weeks after the PN consultation; (2) the effect of APT on the proportion of sex partners treated; (3) the cost effectiveness of APT on long-term sexual and reproductive health outcomes, based on a linked transmission dynamic modelling study; (4) the effects of APT according to sex partner type; (5) whether APT is associated with faster treatment than standard PN; (6)

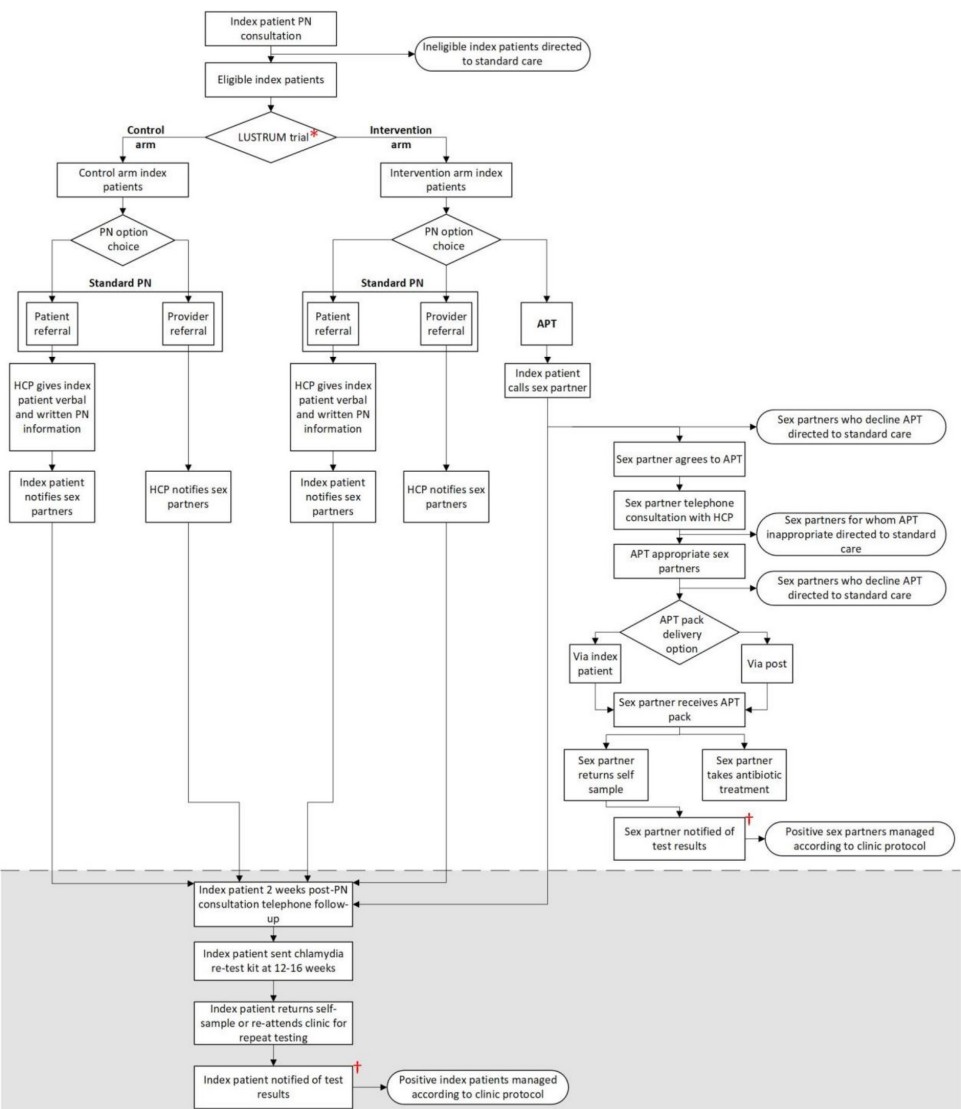

**Figure 1** Index patient and sex partner pathways during control and intervention arms. *Random allocation to intervention or control arm in the first phase of the trial. †Index patients and sex partners are notified of results via two pathways: (1) negative results: individual receives a text from The Doctors Laboratory; (2) positive/equivocal results: individual receives results directly from clinic. APT, accelerated partner therapy; HCP, healthcare professional; LUSTRUM, Limiting Undetected Sexually Transmitted infections to RedUce Morbidity; PN, partner notification.

the effects of APT on transmission of chlamydia at the population level, based on mathematical modelling; and to conduct: (7) a comprehensive process evaluation to understand the experiences of healthcare professionals, patients and sex partners of APT.

## METHODS AND ANALYSIS
### Trial design
LUSTRUM is a cross-over cluster RCT of APT offered as an additional PN method compared with standard PN alone. The APT intervention is offered at the level of the sexual health clinic, with randomisation of each clinic to either intervention or control arm in the first phase of the trial. The trial design is summarised in figure 1.

### Settings
Seventeen NHS (publicly funded, free to access) specialist sexual health clinics (clusters) across England and Scotland with high volumes of positive *C. trachomatis* test results and with contrasting patient demographics. Clinics were selected from those expressing interest, based on numbers of reported chlamydia diagnoses data in the Public Health England Genitourinary Medicine Clinical Activity Dataset for STI surveillance (England) and geographical diversity (Scotland) to create three strata: London, non-London metropolitan 'cities' and non-London urban 'towns'. A full list of study sites is included in the Acknowledgments.

### Eligibility criteria for index patients
Aged 16 years or older, positive test for *C. trachomatis* and/or clinical diagnosis of PID or cervicitis (women) or

non-gonococcal urethritis (NGU) or epididymo-orchitis (men) and report at least one contactable sexual partner in the past 6 months. Index patients with PID, cervicitis, NGU and epididymo-orchitis whose test results are subsequently found to be negative for *C. trachomatis* will not be included in statistical analysis. Exclusion criteria are: groups of patients with more complex PN requirements; co-infection with other STIs including HIV, men who have had sex with men in the past 6 months, and patients who have paid for or who have been paid for sex in the past 6 months. Patients with any clinical, social or other circumstances, such as sexual assault or insufficient English language skills to safely engage in telephone consultations, which emerge during the face-to-face consultation and make APT unsuitable will also be excluded.

### Eligibility criteria for sex partners

Named as a sex partner of the index patient within the appropriate look back period (6 months for chlamydia, 3 months for PID, 1 month for NGU)[29] and selected by the index patient for APT and aged 16 years or older.

### Intervention

The APT intervention is a complex intervention involving index patient, healthcare professional and sex partner, situated within wider clinical care (figure 1). In qualitative research studies to optimise the intervention, we explored the acceptability of individual components of the APT intervention with members of the public, sexual health clinic attenders and healthcare professionals.[30] During the intervention phase, clinics will offer APT as an additional option for eligible patients, alongside standard PN, described in box 1 for index patients, and box 2 and figure 2 for sex partners. Eligible index patients who receive standard PN will receive the same follow-up as those who receive APT. If APT is no longer feasible (eg, sex partner unavailable), standard PN will be offered instead.

### Control

During the control phase, each clinic follows their usual protocols for standard PN. All clinics conduct enhanced patient referral, including verbal information about notifying partners plus additional information (written leaflets or signposting to a website). The healthcare professional will occasionally notify partners on behalf of the patient, if requested. Follow-up telephone calls and repeat testing will be the same as those during the APT phase.

### Outcomes

The primary clinical outcome is the proportion of index patients with a positive test result for *C. trachomatis* 12–16 weeks after the PN consultation. A repeated positive test result after treatment is a proxy for re-infection from an untreated partner, assuming that other reasons for a repeat positive test result are distributed equally between intervention and control arms. This outcome has been used in RCTs of PN interventions seeking to demonstrate prevention of transmission at the level of the individual.[16]

---

**Box 1  Overview of accelerated partner therapy (APT) for index patients**

1. Index patient has partner notification (PN) consultation with healthcare professional (HCP); HCP assesses eligibility for APT.
2. Eligible index patient is offered APT alongside the clinic's other standard PN options. The patient can choose different methods for different partners.
3. Index patient telephones or messages sex partner (with or without the HCP present, according to preference) to offer immediate telephone assessment by the HCP.
4. Index patient waits in clinic while the HCP conducts APT telephone consultation in private with sex partner.
5. Index patient is informed that they will receive a follow-up telephone call in 2 weeks and they will either receive a chlamydia self-sampling postal kit in 12–16 weeks (preferred), or they may re-attend the clinic for testing.
6. Two-week follow-up: research health adviser (RHA) telephones index patient to find out about PN outcomes with partner(s), to remind them of the repeat test, and to invite them to be contacted about taking part in a telephone interview regarding their experiences of APT (process evaluation).
7. 12 weeks: index patient is sent a personalised text reminder about repeat test.
8. 13 weeks: index patient is sent a self-sample kit by The Doctors Laboratory (TDL). Index patient returns self-collected sample or attends clinic for repeat testing. S/he receives results either via text message from TDL (negative results) or using routine clinic systems (positive or equivocal results). Positive results are managed according to routine clinic protocol. If the index patient does not return a self-sample or attend clinic for repeat testing, they receive a personalised text reminder 8 days after the self-sample kit is sent out, followed by a telephone call from the RHA 13 days after the self-sample kit is sent out. Self-samples received >24 weeks post-PN interview are excluded.

---

Secondary clinical outcomes are: (1) the proportion of sex partners treated 2 weeks after the initial PN consultation; (2) numbers of partners treated per index patient; (3) time to partner treatment and (4) number of patients notified per index patient. These outcomes will be ascertained by index patient reports at the 2-week follow-up telephone call with the research health adviser (RHA). Additional secondary outcomes include exploring the acceptability of the APT intervention to index patients and sex partners through qualitative telephone interviews with patients and partners as part of the process evaluation. We will determine rates of STI and HIV testing in sex partners and proportions of positive test results for STIs and HIV among sex partners who return postal self-sampling kits contained in APT packs. We will determine costs associated with the intervention in economic evaluation studies based on costings of the trial. We will record any adverse events using local activity logs at each trial site and report them to the trial sponsor and ethics committee.

### Sample size

The trial is designed to determine superiority of APT as an adjunct to standard PN compared with standard PN. Our

**Box 2   Overview of accelerated partner therapy (APT) for sex partners**

1. Index patient telephones sex partner to inform them about exposure to chlamydia and offer immediate telephone assessment (APT).
2. If sex partner agrees to APT, healthcare professional (HCP) telephones them and conducts a clinical assessment in private. If appropriate, sex partner is offered an APT pack (delivered by the index patient or mailed directly). Sex partners for whom APT is inappropriate or who do not wish to continue with the APT option, will be advised by the HCP to attend clinic for further management. During the same telephone call, HCP invites sex partner to be contacted about taking part in a telephone interview regarding their experiences of APT (process evaluation).
3. Sex partner receives APT pack (figure 2), which contains: antibiotics (either azithromycin or doxycycline, depending on local clinic practice); condoms; information about chlamydia, gonorrhoea, HIV and syphilis; chlamydia and gonorrhoea self-sampling kit, HIV and syphilis self-sampling kit, and information leaflet about how to take a sample (including link to an explanatory online video: lustrum.org. uk/test-and-treat); request form for the sample to be processed by the lab; envelope for return of self-sampling kits and APT pack packaging (envelope or small box, no branding or other identifiable markings, and which fits through standard letterbox).
4. Sex partner completes self-sampling, labels and returns samples for testing.
5. Sex partner takes antibiotic treatment.
6. Sex partner informed of test results by text (negative results) or routine clinic processes; positive results are managed according to routine care.

calculation is based on recruiting an average of 160 index patients per clinic per trial phase from the 17 participating clinical services (total 5440 patients) and a coefficient of variation in the number recruited of 0.5. We expect that

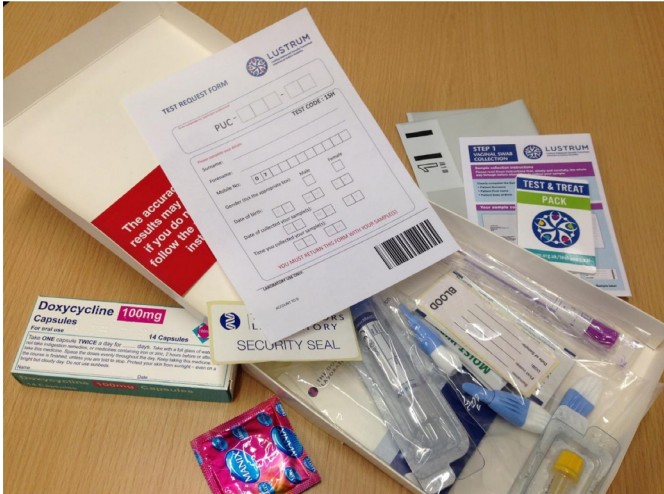

**Figure 2**   Contents of sex partner accelerated partner therapy pack. Pack contains: antibiotics; condoms; Limiting Undetected Sexually Transmitted infections to RedUce Morbidity TEST & TREAT leaflet; vulvo-vaginal swab kit or urine sampling kit; blood sampling kit; instruction leaflet for sampling kits; test request form for sample processing; prepaid return post envelope; security seal sticker; attention card.

50% of recruited patients (80 per clinic per phase and 2720 total) will contribute to the analysis of the primary outcome because we restrict this analysis to patients with *C. trachomatis* detected at baseline. We assume that 60% of index patients will provide a repeat sample for *C. trachomatis* detection and we expect that 10%–25% of patients in the control arm will have *C. trachomatis* detected at the 12–16 weeks follow-up.[16 31 32] This sample size provides 80% power (at the 5% significance level) to detect a fall in *C. trachomatis* positivity from 10% to 5%, and 82% power to detect a fall from 25% to 17%. We consider such reductions would be reasonable if around half the index patients in the intervention phase select APT for some or all their partners, but our calculation does not assume any specific value of APT uptake. A period of 6 months in the control condition and 6 months in the intervention condition should be sufficient to reach this target. Related to our key secondary outcome (the proportion of sex partners treated), for simplicity, we consider power to detect an effect on the proportion of index patients with one or more partner treated. This sample size provides 87% power to detect an increase from 60% in the control arm to 70% in the intervention arm.[17]

Sample size calculations are guided by Giraudeau *et al*[33] and performed as if the trial were a standard cluster RCT with 17 clinics in each arm. The cross-over design in our trial, with each cluster contributing data to both intervention and control conditions, gives more information about the intervention effect and so greater power than stated above. Our calculations assume an intracluster correlation (ICC) of 0.02, which is not based on published data. We believe that our power calculations are likely to be conservative because any loss from a higher ICC will be more than offset by the gain from the cross-over design.

## Recruitment
During the initial PN consultation with the index patient, the healthcare professional will assess eligibility for the study for all patients with a laboratory positive test result for chlamydia or a clinical diagnosis of NGU/epididymo-orchitis (men) or PID/cervicitis (women). During the intervention phase, the healthcare professional will offer eligible patients APT in addition to standard methods of PN. The trial data manager will monitor recruitment and patient uptake of the APT intervention throughout the trial period supplying monthly recruitment figures to the Trial Management Group (TMG) and Data Monitoring Committee (DMC).

## Randomisation/allocation sequence
The random allocation of clinics, to intervention or control arm first, was conducted through random permutation within strata, using computer-generated random numbers in Stata V.15 software. In four strata, the two clinics were part of one NHS trust and were considered as pairs. One stratum contained five clinics from large cities and another contained three clinics from smaller settlements. Given the odd numbers of clinics in these strata,

we used random assignment to determine whether two or three clinics would implement the intervention first in the larger stratum (correspondingly two or one in the smaller stratum to ensure balance). One further clinic, the last (17th) to start recruitment was allocated through simple randomisation, giving a total of nine clinics allocated to implement the intervention first and eight to first follow routine practice. To remove the potential for allocation bias arising because allocation codes were generated after clinics had already been recruited to the trial, one statistician generated the allocation codes and another randomly permuted the clinic names within the strata. Only then, a third person matched the two to reveal the allocations and inform the clinic manager.

## Blinding

Blinding of allocation will not be possible for healthcare professionals, patients or research staff. A statistician blinded to allocation will replicate the analysis of the primary outcome.

## Procedure for collecting data

Healthcare professionals will record their consultations in real time and download data needed for clinical purposes into the patient's clinic record. They will use RELAY, a bespoke data collection tool. RELAY is a secure web-based data collection platform, which incorporates different interfaces allowing specified levels of access to healthcare professionals, RHAs, data manager, trial manager and the research team. It is hosted on secure servers and compliant with NHS data storage requirements. We developed RELAY for the LUSTRUM trial, based on previous versions of data collection tools in our pilot studies of APT.[25 26]

## Statistical analysis

Analysis will be by intention-to-treat. For the primary, and other quantitative outcomes, we will fit random effects models with fixed effects for clinic and intervention condition, together with a random effect to acknowledge the clustering of index patients for each combination of clinic and for each period.[34] The primary and key secondary outcomes are binary, so we will use logistic regression models. The intervention effect will be expressed as an OR with 95% CI. Models for secondary outcomes with outcomes quantified for each sexual contact will include additional random effects for the index patient.

We anticipate loss to follow-up of around 40% of participants, raising potential concerns over bias. Before conducting our analysis of the primary outcome we will analyse predictors of follow-up from among a prespecified list of characteristics of the index patients that will be recorded at baseline and which are considered to be likely predictors of the primary outcome. In the primary analysis of the primary outcome we will include these as covariates in our regression model along with any other key predictors of the outcome. This approach should reduce bias from loss to follow-up but we will also conduct

further sensitivity analyses based on multiple imputation in which we will allow those lost to follow-up to be more, and then less, likely to be chlamydia positive at 3 months than those not lost to follow-up with the same baseline covariates. These sensitivity analyses will follow the approach of Carpenter *et al*,[35] to assess the robustness of our conclusions to different prespecified assumptions about missing data.

## Exclusions from analyses

The intervention is implemented at the clinic level and applied to all eligible patients with service-level, rather than individual, consent. Patients can opt-out of their data being used for research (see online supplementary file 1). Once a clinic has begun the allocated intervention then all eligible patients added to the RELAY web tool will be included in analysis, following the spirit of an intention-to-treat analysis and the all-randomised population. However, in the unlikely event of a substantial period in which any clinic is subsequently unable to implement the intervention for any reason then we will also conduct an analysis that only includes patients attending the clinic while it is able to offer the intervention, in the spirit of a per-protocol analysis and a protocol compliant population. The inclusion of patients in analysis will not be influenced by whether the patient does or does not take up the offer of APT.

Patients who present with conditions managed at first attendance as presumptive urogenital chlamydia but who subsequently are found to have negative chlamydia tests will be excluded from analyses.

## Trial timeline

There will be a 4-month run in period (July–October 2018), consisting of rolling clinic set-up including training for healthcare professionals and a period of at least 2 weeks of baseline data collection when healthcare professionals will use RELAY to record standard PN data. Research staff will monitor completeness of baseline data collection. The data will be used at the end of the trial to help interpret the trial findings. Then half of the clinics will enter intervention phase while the other half enter control phase, according to the randomisation schedule. The trial is planned to start 22 October 2018.

At the end of the first 6-month trial phase (October 2018–April 2019) there will be a 2-week washout period where clinics will not offer APT to patients and follow their standard PN practice procedures. Clinics which start with the control phase will also observe the 2-week washout to align the trial time periods. Then clinics cross over to the opposite arm (intervention or control) for phase two (for 6 months, May–November 2019); patient recruitment is planned to end 17 November 2019. The total duration of the trial will be 19 months, allowing for a 3-month follow-up period to enable outcome data collection to be completed for all patients in the second trial phase. Clinics which do not start phase one of the trial in November 2018 will complete recruitment in November

2019 and trial phases will be condensed. The LUSTRUM programme as a whole runs from April 2016 to March 2021.

## Process evaluation

We will conduct an integrated process evaluation, which involves a suite of substudies with index patients, sex partners and healthcare professionals. In brief these studies include: (1) qualitative telephone interviews with a sample of index patients who accepted APT for one or more sex partner, which will take place up to 8 weeks after their first PN consultation; (2) qualitative telephone interviews with a sample of sex partners who received APT, up to 8 weeks after their APT consultation; (3) qualitative telephone interviews with healthcare professionals who delivered APT, which take place up to 16 weeks after the clinic offers APT during the intervention phase; (4) focus groups with healthcare professionals to explore the context, extent and fidelity of the intervention delivery, which will take place up to 16 weeks after the clinic offers APT during the intervention phase.

All potential participants will receive the relevant participant information sheet and the researcher will discuss this information with participants before seeking informed consent to participate (see online supplementary files 2–5). All telephone interviews will be audio-recorded with informed verbal consent prior to participation, transcribed verbatim and analysed using interpretative phenomenological analysis[36] to understand lived experiences of using APT. The expected duration of telephone interviews is 30–60 min. Focus groups will be audio-recorded with informed written consent prior to participation, transcribed verbatim and data will be analysed using the Framework approach.[37] Data collected will assess intervention fidelity, healthcare professionals' APT training (the behaviour change techniques they received) and their experience of APT delivery (the behaviour change techniques they themselves delivered) in relation to a range of feasibility issues such as impact on clinic flow and wider systemic and cultural issues.

## Health economics evaluation

The aim of the economic evaluation is to determine the relative cost effectiveness of APT compared with standard PN. If APT reduces the number of re-infected cases, there will be cost implications for the healthcare sector and for society as a whole because of effects on the incidence of complications. Cost effectiveness will be determined in two ways using two separate economic analyses which will explore the costs and consequences of the alternative interventions and report results in terms of incremental cost effectiveness ratios if applicable. The two analyses are: (1) within trial economic evaluation which will report the terms of natural units including *cost per case of re-infection avoided* and (2) model-based economic evaluation which will report results in terms of cost per QALY (Quality-Adjusted Life Year) based on new data from recent research.[38]

The within trial evaluation will use only data collected within the trial and will draw on data collected through provision of APT up until 4 months postintervention when the index cases are re-tested. The evaluation will adopt the perspective of the NHS and will be based on an outcome of cost per case of re-infection avoided. It is acknowledged that this is an intermediate outcome. We will build on the experience of exploring the cost effectiveness of alternative versions of APT in previous exploratory and pilot studies.[25 26]

The trial will not capture the effects of APT on the transmission of chlamydia, or on female reproductive tract complications, such as PID. As for previous economic evaluations of chlamydia screening interventions,[39] we will develop a dynamic model of the epidemiology of *C. trachomatis* transmission, the APT intervention and its outcomes, which will use data from the trial. The model-based economic evaluation and deterministic transmission dynamic model on which the economic evaluation will be based, will be reported in detail in a separate publication.

## Trial monitoring

The TMG is responsible for the day-to-day design, delivery and management of the trial.

The Trial Steering Committee (TSC) is an executive and independent body, providing overall supervision of the trial and ensuring that it is being conducted in accordance with the principles of good clinical practice and the relevant regulations. The TSC will approve the trial protocol and any protocol amendments and provide advice to investigators on all aspects of the trial. The TSC will monitor trial progress including recruitment, data completeness, and losses to follow-up and ensure that there are no major and/or unexplained deviations from the trial protocol.

The DMC will be responsible for reviewing interim trial data and reporting back to the sponsor on the future management of the trial. The DMC will use data accumulated throughout the trial to inform guidance and recommendations for the continued ethical conduct of the trial. The APT intervention is a short, one-off intervention provided to patients as an additional PN option which complements existing standard care practices. As such, the risk of APT to patients is minimal. We will not conduct a formal interim analysis.

We will collect reports of adverse events relating to either the intervention or participation in the trial, which will be reviewed by the TSC and reported to the sponsor and ethics committee as appropriate. The TSC could recommend the trial be stopped if members are sufficiently concerned about these events. The trial will not be stopped early due to demonstration of efficacy, harm or futility in the primary outcome.

## Patient and public involvement

The LUSTRUM PPI (Patient and Public Involvement) Group consists of 27 lay people, with a broad mix of

demographic characteristics and a range of sexual health experiences. The group were first involved in the design of the research at the grant application stage. Their opinions guided the trial protocol and topic guides for the process evaluation interviews. The trial is conducted by qualified healthcare professionals and, as such, it is not feasible for the group to undertake any aspects of the data collection. The group strongly supported the service-level consent design which reduces the burden of participating in the research. Lay summaries of our findings will be circulated to the group to gain their perspective and feedback on our results; this may inform further analytical decisions and inputs.

## ETHICS AND DISSEMINATION

As this is a pragmatic cluster randomised trial, we will seek consent for trial participation from lead clinicians at participating clinics (service-level) and will not seek individual informed consent from index patients other than for the process evaluation studies. Following Weijer *et al*,[40] we believe that APT is a complex, 'low-risk' healthcare delivery intervention. APT is offered in addition to standard PN and operationalised as a supplement to usual care, thus index patients have the choice of taking up APT or not. It is widely accepted that individual consent may not be essential in such trials,[41] in which the situation is analogous to the introduction of changed processes in routine services.[31] Notably, individual level consent is thought to have contributed to low recruitment numbers in a previous study of APT.[26] Individual informed consent will be sought from all participants in the process evaluation.

Study endpoints, whether negative or positive, will be reported and disseminated through the following channels. (1) Research findings will be published in journals with open access within 6 months of publication. (2) Research findings will be presented at UK and international meetings orally or via posters. (3) A report of the findings (in the form of the funder's final report) will be freely available on both the funder's website and the LUSTRUM website after publication in scientific journals. A link to the report will be circulated to stakeholders, collaborators and participating organisations, together with lay summaries of the study on the LUSTRUM website. (4) We will make presentations to community groups and clinic user groups as appropriate. (5) We will make continued use of social media channels to report findings after publication in scientific journals, in order to increase the 'reach' of our findings to the public. (6. The anonymised participant level dataset, and statistical code for generating the results will be archived in a data repository after publication of the final report.

## Author affiliations
[1] School of Health and Life Sciences, Glasgow Caledonian University, Glasgow, UK
[2] Institute for Global Health, UCL, London, UK
[3] Institute of Social and Preventive Medicine, University of Bern, Bern, Switzerland
[4] Institute of Health and Wellbeing, University of Glasgow, Glasgow, UK
[5] Health Economics Unit, University of Birmingham, Birmingham, UK
[6] Development Media International CIC, London, UK
[7] Western Sussex Hospitals NHS Foundation Trust, Worthing, UK
[8] NHS Greater Glasgow and Clyde, Glasgow, UK
[9] Public Health England, London, UK
[10] All East Sexual Health, Barts Health NHS Trust, London, UK
[11] Brighton and Sussex Medical School, East Sussex, UK

**Acknowledgements** The authors would like to thank the following: (1) Trial Steering Committee: Simon Barton (chair), Robbie Currie, David Crundwell, Artemis Koukounari, Lynis Lewis, Alec Miners, Emmanuel Rollings-Kamara, Rachel Shaw, Emma Thompson. (2) Data Monitoring Committee: Simon Barton, David Crundwell, Rebecca Turner, Artemis Koukounari. (3) All members of the LUSTRUM Patient and Public Involvement Group. (4) Participating centres: Barking, Havering and Redbridge University Hospitals NHS Trust (A. Umaipalan), Barts Health NHS Trust (V. Apea), Buckinghamshire Healthcare NHS Trust (J. Sherrard), Chelsea and Westminster Hospital NHS Foundation Trust (C. Evans), Croydon Health Services NHS Trust (D. Phillips), Manchester University NHS Foundation Trust (G. Schembri), Midlands Partnership NHS Foundation Trust (J. Dhar), NHS Greater Glasgow and Clyde (R. Nandwani), Northamptonshire Healthcare NHS Foundation Trust (S. Herbert), Royal Berkshire NHS Foundation Trust (F. Chen), Royal Bournemouth and Christchurch Hospitals NHS Foundation Trust (K. Schroeder), Solent NHS Trust (R. Patel), University Hospitals Birmingham NHS Foundation Trust (J. Ross). The authors would also like to thank epiGenesys who developed the RELAY software; Soazig Clifton (support with ethics submission); Jo Gibbs (support on RELAY development); Kay Musonda (administrative support) and the staff, management and patients involved in the study at all the participating sexual health clinics.

**Contributors** CSE is chief investigator of the study. CSE, JAC, NL, TR, AnC, AMJ, CHM, JS, MS, PF, RN and SW conceived the study and secured funding. They are responsible for the planning and delivery of the study. PF leads the process evaluation; TR leads on the health economics evaluation and NL leads on the mathematical modelling. AnC is the trial statistician. ARH, FM and MWO are responsible for coordination of the study. ARH, FM, MWO, KP, C(D)O, SB, AIC, TM, MP, GV and AT are responsible for operationalisation of the study procedures. FM, MWO, KP, C(D)O, MP and GV will contribute to data collection and analysis. ARH, SB and AIC will contribute to data collection. CA and NL are responsible for the mathematical modelling. AT is responsible for trial data management and will contribute to data analysis. All authors contributed to the development of the study design and establishment of procedures. CSE led on preparing the manuscript. All authors critically reviewed and approved the final version.

**Funding** This work presents independent research funded by the National Institute for Health Research (NIHR) under its Programme Grants for Applied Research Programme (reference number RP-PG-0614-20009).

**Disclaimer** The views expressed are those of the author(s) and not necessarily those of the NHS, the NIHR or the Department of Health. The funders had no role in study design, collection, management, analysis and interpretation of data; writing of the report and the decision to submit the report for publication.Trial sponsor: Noclor, Central and North West London NHS Trust. The sponsor had no role in study design, collection, management, analysis and interpretation of data; writing of the report and the decision to submit the report for publication.

**Competing interests** None declared.

**Patient and public involvement** Patients and/or the public were involved in the design, or conduct, or reporting, or dissemination plans of this research. Refer to the Methods section for further details.

**Patient consent for publication** Not required.

**Ethics approval** Ethical approval from London—Chelsea Research Ethics Committee (18/LO/0773).

**Provenance and peer review** Not commissioned; externally peer reviewed.

ORCID iDs
Claudia S Estcourt http://orcid.org/0000-0001-5523-5630
Nicola Low http://orcid.org/0000-0003-4817-8986
Fiona Mapp http://orcid.org/0000-0003-0733-6036
Jackie A Cassell http://orcid.org/0000-0003-0777-0385

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
