## [Reviewer comments · BMJ Open]

ARTICLE DETAILS

TITLE (PROVISIONAL)	Accelerated Partner Therapy (APT) partner notification for people with Chlamydia trachomatis: protocol for the Limiting Undetected Sexually Transmitted infections to Reduce Morbidity (LUSTRUM) APT cross-over cluster randomised controlled trial
AUTHORS	Estcourt, Claudia; Howarth, Alison; Copas, Andrew; Low, Nicola; Mapp, Fiona; Woode Owusu, Melvina; Flowers, Paul; Roberts, Tracy; Mercer, Catherine; Wayal, Sonali; Symonds, Merle; Nandwani, Rak; Saunders, John; Johnson, Anne; Pothoulaki, Maria; Althaus, Christian; Pickering, Karen; McKinnon, Tamsin; Brice, Susannah; Comer, Alex; Tostevin, Anna; Ogwulu, Chidubem (Duby); Vojt, Gabriele; Cassell, Jackie;

VERSION 1 – REVIEW

REVIEWER	Jesse Clark UCLA Geffen School of Medicine
REVIEW RETURNED	07-Nov-2019

GENERAL COMMENTS	The protocol is well written, providing an excellent review of the topic of partner management and a detailed overview of this important clinical trial of APT for Chlamydia infection among heterosexual STI clinic patients in the UK. I have no criticisms or comments to be addressed.
--

REVIEWER	Okeoma Mmeje University of Michigan, Ann Arbor, Michigan United States of America
REVIEW RETURNED	10-Jan-2020

GENERAL COMMENTS	The LUSTRUM study protocol is very well written and addresses all the possible challenges and implications that may be encountered. It offers an innovative and thorough approach to partner notification that hopefully circumvents some of the traditional barriers that healthcare providers often cite as to why they limit expedited partner therapy (US perspective). I look forward to learning about the study findings and its clinical implications in England and other countries that employ partner notification systems for STIs.
---